



# Note on the directional properties of meter-scale gravity waves

Charles Peureux[1], Alvise Benetazzo[2], and Fabrice Ardhuin[1]

[1]Laboratoire d'Océanographie Physique et Spatiale, Univ. Brest, CNRS, Ifremer, IRD, 29200 Plouzané, France
[2]Institute of Marine Sciences, Italian National Research Council, Venice, Italy

*Correspondence to:* Charles Peureux (charles.peureux@univ-brest.fr)

**Abstract.** The directional distribution of the energy of young waves is bimodal for frequencies above twice the peak frequency, and that distribution can be obscured by the presence of bound waves. Here we analyze in detail a typical case measured with a peak frequency $f_p = 0.18\,\mathrm{Hz}$ and a wind speed of $10.7\,\mathrm{m \cdot s^{-1}}$. The directional distribution for a given wavenumber is nearly symmetric, with the separation of the two lobes of the directional distribution growing with frequency, reaching 150° at 35 times the peak wave number $k_p$ and increasing up to 45 $k_p$. When considering only free waves, the lobe ratio, the ratio of oblique peak energy density over energy in the wind direction, increases linearly with the non-dimensional wavenumber $k/k_p$, up to a value of 6 at $k/k_p \simeq 22$, possibly more for shorter components. These observations extend to shorter components previous measurements, and have important consequences for wave properties sensitive to the directional distribution, such as surface slopes, Stokes drift or microseism sources.

## 1  Introduction

Directional properties of waves shorter than the dominant scale play a very important role in many aspects that range from air-sea momentum fluxes (Plant, 1982) to remote sensing, surface drift (Ardhuin et al., 2009) and underwater acoustics (Duennebier et al., 2012). In a landmark paper, Munk (2009) proposed an interpretation of directional wave properties using an analogy with ship wakes based on slope statistics derived from the very large satellite dataset of Bréon and Henriot (2006). As he put it, the dataset *says nothing about time and space scales* because they are integrated across all wave scales. Munk further challenged us all, *I look forward to intensive sea-going experiments over the next few years demolishing the proposed interpretations*. We thus went out to sea with the objective of resolving space and time scales, and providing further constraints on the wave properties.

Previous time-resolved measurements of ocean waves have clearly established a prevalence of directional bimodality at frequencies above twice the peak frequency $f_p$, using in situ array (Young et al., 1995; Long and Resio, 2007), and buoy data (Ewans, 1998; Wang and Hwang, 2001). These were confirmed by airborne remote sensing techniques used by Hwang et al. (2000), and Romero and Melville (2010). All the resolved wavenumber spectra have been limited to $f/f_p < 4$. Numerical modelling by Banner and Young (1994) suggests that the bimodality is caused by the nonlinear cascade of wave energy from





dominant to high frequencies. The model results of Gagnaire-Renou et al. (2010, their figure 18) show that there may or may not be a transition back to unimodal directional distributions, somewhere below $f/f_p = 10$, depending on the parameterizations of wave generation and dissipation.

The directional distribution of backscatter clearly shows that the directional wave spectrum is unimodal above $6\,\mathrm{cm}$ wave-
length in the gravity-capillary range (see the review in Elfouhaily et al., 1997). Recent backscatter data in L-band presented by Yueh et al. (2013) show a larger cross-wind than down-wind backscatter, consistent with a bimodal distribution at scales around $1\mathrm{m}$ wavelength, at least for wind speeds around $5\,\mathrm{m\cdot s^{-1}}$.

As shown by Leckler et al. (2015), stereo-video imagery is capable of resolving these waves and give information on the time and space scales needed to interpret integrated wave parameters such as surface slope. Here we extend these previous
measurements to shorter waves and discuss their implications.

The data and analysis methods are presented in section 2. Directional distributions and bimodality are described in section 3. Discussions and conclusions follow in section 4.

## 2   Wave measurements and spectral analysis

### 2.1   Stereo processing

We have chosen one typical stereo record with dominant waves longer than those described in Leckler et al. (2015). It was acquired on 10 March 2014, starting at 09:40 UTC, from the Acqua Alta oceanographic research platform, $15\,\mathrm{km}$ offshore of Venice, Italy, in the northern Adriatic Sea. The mean water depth there is approximately $d = 17\,\mathrm{m}$. The experimental setup has been described in detail in Benetazzo et al. (2015). It is made of two digital cameras mounted on a horizontal bar, properly synchronized and calibrated. The cameras are located $d = 12.5\,\mathrm{m}$ above the mean sea level. The stereo device is pointing in a
direction oriented $46°$ clockwise from geographical North, i.e. looking to the North-East. The cameras elevation angle is $50°$. This record is 30 minutes long, and uses a $15\,\mathrm{Hz}$ sampling rate.

In the following all variables use the meteorological convention, namely the directions are directions from which wave, wind and current come from. Provenance directions, unless otherwise specified, are measured anticlockwise from the direction along the bar, i.e. $136°$ clockwise from geographical North.
The mean wind speed measured at $10\,\mathrm{m}$ above sea level is 10.7 m/s, with mean direction $\theta_U = 77°$ (north-easterly). The significant wave height estimated from the stereo system is $H_{m0} = 1.33\,\mathrm{m}$, with peak frequency $f_p = 0.185\,\mathrm{Hz}$, corresponding to a dominant wavelength of the order of $45\,\mathrm{m}$. We note that wave gauges on the platform give independent measurements of $H_{m0} = 1.36\,\mathrm{m}$ and $f_p = 0.189\,\mathrm{Hz}$. Dominant waves and shorter components of the wave spectrum can be considered deep water waves. An Acoustic Doppler Current Profiler (ADCP) deployed at the sea floor provides measurements of the horizontal
current vector with a vertical resolution of $1\,\mathrm{m}$.

The raw video images are processed into a three-dimensional surface elevation matrix $\zeta(x,y,t)$ following the method of Benetazzo et al. (2015). A local cartesian reference frame is defined, in which the surface elevation is reconstructed, with horizontal axes $x$ and $y$. By convention, the cameras look direction is the $y$ axis, increasing away from the cameras, and the $x$

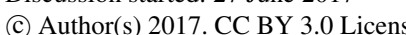



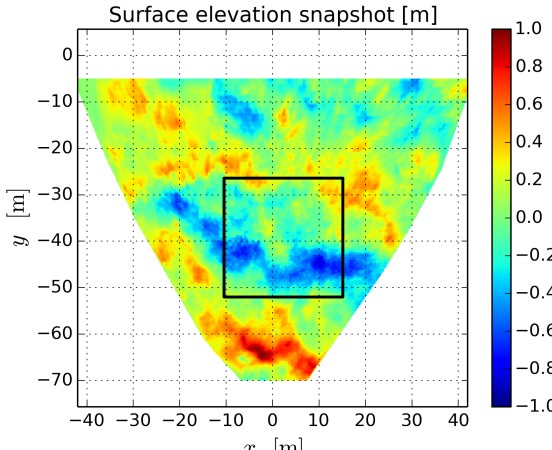

**Figure 1.** Stereo-video reconstructed sea-surface elevation matrix snapshot and sea-surface area used for spectrum calculations (black square).

axis is perpendicular, increasing towards the right of the cameras. The sea surface is discretized with a pixel size $\Delta x = \Delta y = 20\,\mathrm{cm}$. A snapshot of the reconstructed sea-surface elevation map is presented on Fig. 1. We have selected a 25.6 by 25.6 m area for Fourier analysis, delimited by a black square. Its location, close to the cameras, is chosen to minimize errors in the estimate of the surface elevation. These errors increase with increasing distance from the cameras, and are dominated by the

quantization error (Benetazzo, 2006).

All our analysis is based on a three-dimensional power spectral density of this data (Fig. 2 and 3). This is obtained by applying a Hann spatiotemporal window with 50% overlap in time, and averaging the spectra in time following Welch (1967). The frequency resolution is $\Delta f = 0.015\,\mathrm{Hz}$. The double-sided cartesian spectrum $E(k_x, k_y, f)$ is normalized so that

$$E = \iiint dk_x dk_y df\, E(k_x, k_y, f) \tag{1}$$

is the variance of the surface elevation.

The polar spectrum is more convenient for the study of directional distributions and for working at given wavenumber. The single-sided polar spectrum is

$$E(k, \theta, f) = 2k E(k_x, k_y, f), \tag{2}$$

where $k = (k_x^2 + k_y^2)^{1/2}$ and $\theta = \arctan(k_y, k_x)$ is the waves provenance direction. For convenience, we use a regular polar

grid, which resolution is set to $\Delta k = 0.17\,\mathrm{rad} \cdot \mathrm{m}^{-1}$ and $\Delta\theta = 1°$.





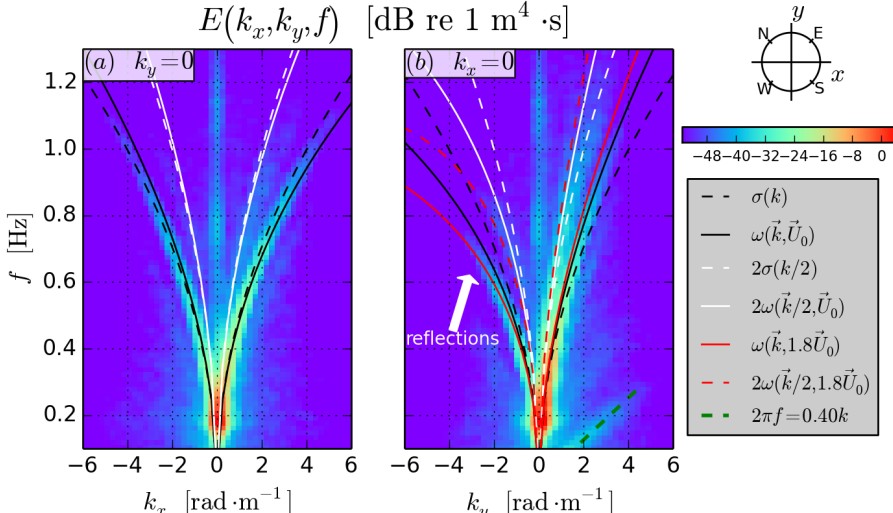

**Figure 2.** Frequency-wavenumber surface elevation spectrum at Acqua Alta on March 10$^\mathrm{th}$ 2014 in dB and various dispersion lines. Spectrum along the cross-look (a) and look direction (b).

## 2.2 General properties of the 3D spectrum

The surface elevation spectrum can be interpreted as the distribution of wave energy, which can generally be divided into free and bound waves,

$$E(\boldsymbol{k}, f) = E_\mathrm{free}(\boldsymbol{k}, f) + E_\mathrm{bound}(\boldsymbol{k}, f). \tag{3}$$

5    Free waves have a relation between wavenumber and frequency that closely follows the linear dispersion relation. In the presence of a horizontally homogeneous and stationary current vertical profile $u(z)$, and in the limit of small wave steepness, this dispersion relation is given by Stewart and Joy (1974)

$$\omega(\boldsymbol{k}, \boldsymbol{U}) = \sigma(k) + \boldsymbol{k} \cdot \boldsymbol{U}(k), \tag{4}$$

where

10   $$\sigma(k) = \sqrt{gk\tanh(kd)} \tag{5}$$

and the effective current $\boldsymbol{U}(k)$ is approximated by a weighted integral of the eulerian current $u(z)$ over the water column

$$U(k) = 2k \int_{-\infty}^{0} u(z)e^{2kz}dz. \tag{6}$$

Here we have assumed that the current has a constant direction $\alpha$ at all depths. Moreover, Eq. (6) holds only for linear waves, *i.e.* waves for which hydrodynamic nonlinearities have been neglected, although free waves may encompass some weakly





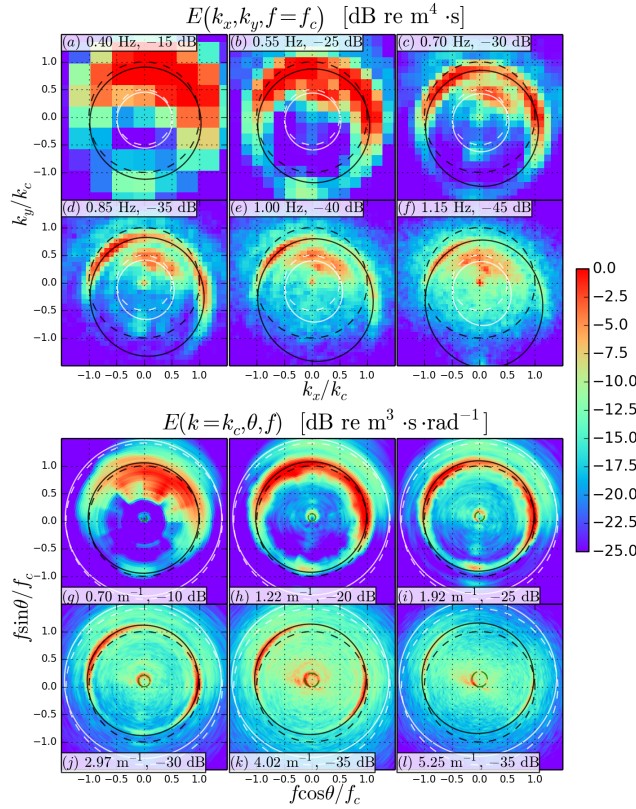

**Figure 3.** Surface elevation spectrum at Acqua Alta (continued). (a) to (f): cartesian spectrum at constant frequencies $f_c$. (g) to (l): cartesian spectrum interpolated into polar coordinates at constant wavenumbers $k_c = \kappa(f_c, 0)$. The reference levels are indicated in the text boxes. See legend on Fig. 2 for the various lines meaning.

nonlinear contributions - see Leckler et al. (2015) and Janssen (2009). The depth weighting in the integral of Eq. (6) gives a stronger influence of surface currents to shorter wave components. In practice, waves with wavenumber $k$ feel the surface integrated current over a depth $\sim 1/k$. For convenience, the inverse function providing the wavenumber as a function of frequency and direction will be denoted $\kappa$ in the following, namely by definition, if

$$k = \kappa(\boldsymbol{f}, \boldsymbol{U}), \tag{7}$$

then

$$2\pi f = \omega(\boldsymbol{k}, \boldsymbol{U}), \tag{8}$$

where $\boldsymbol{f} = [f\cos(\theta), f\sin(\theta)]$.

Once the effective current (6) is known, the location of free waves in the $(\boldsymbol{k}, f)$ plane can be deduced from Eq. (4), which relates the radian frequency $2\pi f$ to the wave vector $\boldsymbol{k}$. It is represented on Fig. 2 and 3 by a black solid line. The addition of a



current is necessary to fit the observations of energy distribution. The free modes bimodality is clearly visible, *i.e.* the fact that two energy patches detach progressively from a main direction as the waves scale decreases.

Bound waves are dominated by the second-order interaction of free components with wavenumbers $\boldsymbol{k}_1$ and $\boldsymbol{k}_2$. The sum interaction gives waves of wavenumber $\boldsymbol{k} = \boldsymbol{k}_1 + \boldsymbol{k}_2$, and frequency $\omega = \omega(\boldsymbol{k}_1) + \omega(\boldsymbol{k}_2)$, with an energy $E_{\text{sum}}$. The differ-

ence interaction gives $\boldsymbol{k} = \boldsymbol{k}_1 + \boldsymbol{k}_2$ and $\omega = |\omega(\boldsymbol{k}_1) - \omega(\boldsymbol{k}_2)|$, $E_{\text{diff}}$. These two kinds of interactions have themselves distinct signatures in the surface elevation spectrum, namely

$$E_{\text{bound}}(\boldsymbol{k}, f) = E_{\text{sum}}(\boldsymbol{k}, f) + E_{\text{diff}}(\boldsymbol{k}, f). \tag{9}$$

At given propagation direction, the sum interaction is found at frequencies higher than the dispersion surface, while the difference interaction components are found at lower frequencies (Leckler et al., 2015; Krogstad and Trulsen, 2010). $E_{\text{bound}}$

can be deduced from $E_{\text{free}}$ (Hasselmann, 1962). More specifically, for a narrow spectrum, the sum interaction component is characterized by a signature in the $(\boldsymbol{k}, f)$ plane (Senet et al., 2001)

$$2\pi f = 2\omega(\boldsymbol{k}/2, \boldsymbol{U}), \tag{10}$$

also referred as first harmonic. The latter corresponds to sum interactions of free waves traveling in the same direction, with same frequency and propagation direction, for which the interaction cross section is highest (Aubourg and Mordant, 2015).

This curve is represented on Fig. 2 and 3 by a white solid line. Its equivalent without current is also plotted as a white dashed line. Nonlinear components do not exhibit the same directionality as linear waves in general, especially from snapshots at constant frequency. In this case, the harmonic peaks in the dominant wave direction (see Fig. 3, panels b–f), although this is not the only possible behavior.

We also note that waves that are probably reflected by the platform legs are present, as shown by a white arrow on Fig. 2b,

and their energy decreases with increasing distance. When interpreted as plane waves, the reflected components appear slightly off the dispersion relation of the incident waves. Fitting the current for the incident waves gives $U \simeq 0.22\,\text{m}\cdot\text{s}^{-1}$, whereas a fit for the reflected components only would give a current velocity of $0.4\,\text{m}\cdot\text{s}^{-1}$.

Finally, there are other spectral features that do not correspond to surface waves which we shall call noise. We distinguish four kinds of noise. Firstly, a background noise is present below $-50\,\text{dB}$ particularly visible on Fig. 3d–f and j–l. This noise

practically limits the use of stereo-video to $k < 8\,\text{rad}\cdot\text{m}^{-1}$. Secondly, some energy propagates with a speed of $0.4\,\text{m}\cdot\text{s}^{-1}$ along the look direction and at slower speeds for other directions (green dashed lines on Fig. 2b, and 3i–n). For $k = 2\,\text{rad}\cdot\text{m}^{-1}$ this noise amplitude is comparable in magnitude to the free waves signature and is distributed around a surface of the type $2\pi f = 0.4\sin^2(\theta)k$, for $\theta \in [0; \pi[$ only. It could be associated with the difference interaction between incident and reflected wavenumbers.

Besides these noises, uncertainties in the spectral densities are caused by the poor spectral resolution close to $k = 0$, and quantization error noise, mostly for $k > 7.5\,\text{rad}\cdot\text{m}^{-1}$ and in the look direction (Benetazzo, 2006). We thus exclude from our



analysis the spectral components for which any of the following conditions is met

$$f \leq \Delta f \tag{11}$$

$$f > 1.4\,\mathrm{Hz} \tag{12}$$

$$k \leq \Delta k \tag{13}$$

$$k > 7.5\,\mathrm{rad \cdot m^{-1}} \tag{14}$$

$$2\pi f[\mathrm{Hz}] < 1.1 \times 0.4 \times \sin^2(\theta)k[\mathrm{rad \cdot m^{-1}}] \tag{15}$$

Outside of these components the spectrum is separated into free and bound components. This uses a determination of the effective current that is discussed in the Appendix. Identifying the free wave energy as that close to the linear dispersion relation, the bound components are defined as the rest,

$$E_{\mathrm{bound}}(k,\theta) = E(k,\theta) - E_{\mathrm{free}}(k,\theta), \tag{16}$$

and the same is done for the frequency-direction spectrum.

## 3 Directional properties of free waves

The spectrum of free waves $E_{\mathrm{free}}(k,\theta)$ is clearly bimodal for $k > 4k_p$. Bimodal energy distributions can be characterized from the knowledge of the position and height of the energy peaks. The processing starts from the radially integrated directional distributions, both at given frequency $E_{\mathrm{free}}(\theta) = \int dk\, E_{\mathrm{free}}(k,\theta)$ and wavenumber $E_{\mathrm{free}}(\theta) = \int df\, E_{\mathrm{free}}(f,\theta)$ (see Fig. 4). The same processing is performed on bound waves, obtained from Eq. (16). Nonlinearities are found to stand for a significant proportion of the overall energy at such wave scales (62 and $64\,\%$ on panels b and d respectively), but only the free waves are bimodal. The contributions of the sum and difference interactions are also indicated. Directional distributions are centered on the spectral mean direction of wave propagation $\theta_m = 68°$ from Benetazzo et al. (2015).

As the directional distributions are noisy, they need to be fitted by an appropriate shape function. Inspired from Ewans (1998), the double pseudo Voigt function with positive floor has been chosen. The fit is performed using the Python lmfit package (Newville et al., 2014). The double pseudo-Voigt function allows for more various curve shapes than only two gaussian beams, with its 9 degrees of freedom, when the lorentzian fraction $\alpha$ is nonzero,

$$E_{\mathrm{fit}}(\theta) = C^{\mathrm{st}} + f(\theta; A_1, \mu_1, \sigma_1, \alpha_1) + f(\theta; A_2, \mu_2, \sigma_2, \alpha_2), \tag{17}$$

where

$$f(\theta; A, \mu, \sigma, \alpha) = \frac{(1-\alpha)A}{\sigma_g\sqrt{2\pi}}e^{-(\theta-\mu)^2/2\sigma_g^2} + \frac{\alpha A}{\pi}\frac{\sigma}{(\theta-\mu)^2+\sigma^2}, \tag{18}$$

with $\sigma_g = \sigma/\sqrt{2\ln 2}$. The use of a double Voigt profile does not strictly ensure a smooth periodic distribution (around $\theta = \theta_m \pm \pi$). However, in practice, due to the relative directional narrowness of the bimodal profiles, the constant energy floor is


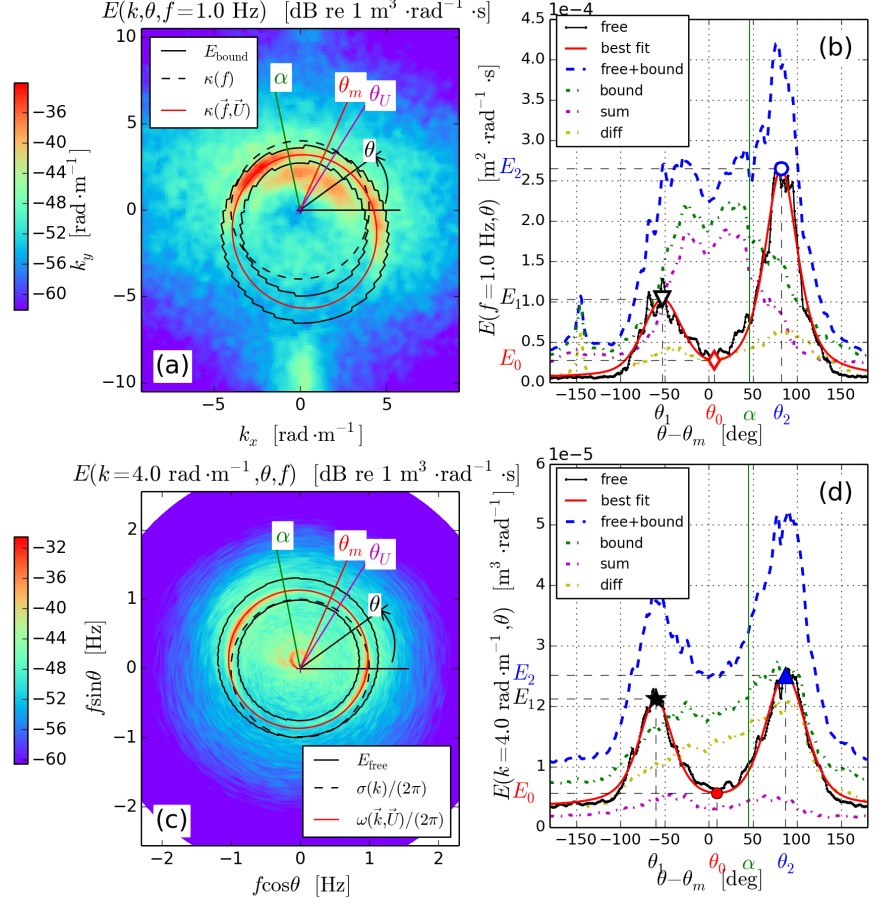

**Figure 4.** Free waves extraction (a and c) and corresponding directional distributions (b and d). Left panels: semi-automatic extraction of free waves, with $\alpha$ the current direction, $\theta_m$ the spectral mean direction of wave propagation and $\theta_U$ the wind direction. Right panels: Directional distributions of free waves (solid line), with fits and the various nonlinear contributions.

quickly reached away from the mean wave propagation direction. Bimodality can then be characterized using a set of three remarkable points in the double Voigt profile (see Fig. 4b and d) (Wang and Hwang, 2001), *i.e.* the central minimum $(\theta_0, E_0)$ and the two peaks $(\theta_1, E_1)$ and $(\theta_2, E_2)$, with $\theta_1 < \theta_2$.

## 4   Results

5   The present case bimodality is characterized by plotting the positions of the two peaks and the so-called lobe ratios as a function of normalized wavenumber $k/k_p$ (see Fig. 5). Full markers (triangles, circles and stars) correspond to estimates from constant wavenumber snapshots while empty markers (disks, diamonds and upside down triangles) correspond to estimates from constant frequency snapshots. For the latter, the $x$-axis is $\kappa(f)/k_p$ - see Eq. (7). Bimodal profiles are first detected at



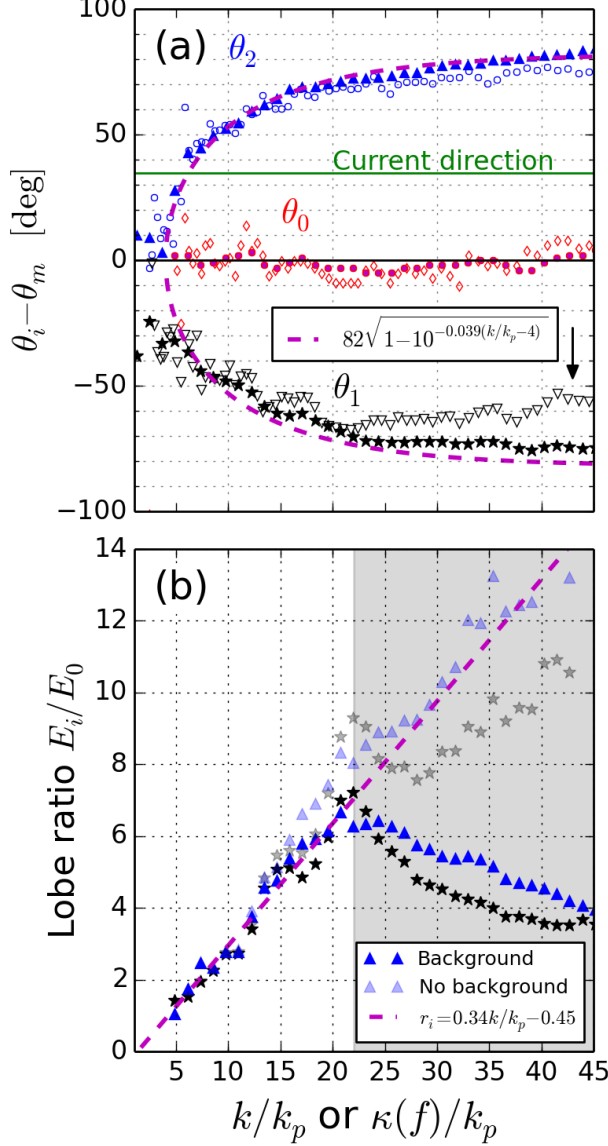

**Figure 5.** Bimodal directional distributions characteristics from stereo-video as a function of normalized wavenumber. (a): peak positions. (b): lobe ratios. Empty markers correspond to constant frequency estimates and full markers to constant wavenumber estimates (see Fig.4 for definitions).

$f = 0.43\,\mathrm{Hz}$ and $k = 0.7\,\mathrm{rad \cdot m^{-1}}$, corresponding approximately to $k/k_p = 4$. The previously mentioned direction $\theta_m$ is the best compromise for centering the bimodality. An empirical parametrization is found for the constant wavenumber estimates





of the directional distributions, that is

$$(\theta - \theta_m)\,[°] = 82\sqrt{1 - 10^{-a(k/k_p - 4)}}. \tag{19}$$

where the value $a = 0.039$ was found after a least squares fit of constant wavenumber data points in the range $8 < k/k_p < 40$. This parametrization fits most of the measurements, except the position of the peak furthest from the current direction ($\theta_1$),

particularly for the estimates from constant frequency snapshots above $k/k_p = 22$ (see black arrow on Fig.5). At this location, the peak is progressively moved towards the center of the directional distribution.

The lobe ratios $r_i$ are conventionally defined as the ratios of the energy of each peak of the bimodal directional distribution to the one of the central minimum (Wang and Hwang, 2001), namely

$$r_i = \frac{E_i}{E_0}, \; i = 1, 2. \tag{20}$$

We can note that they are particularly sensitive to the background wave spectrum encompassed in the constant term of the fitting function (17), which can arise from actual surface waves or noise, particularly when surface waves are not resolved anymore. The lobe ratios of the current record are plotted on Fig. 5b, from estimates at constant wavenumber only, with and without this background term. The overall tendency consists in their linear and symmetric increase at intermediate wave scales, until $k/k_p \simeq 22$. As for peak positions on Fig. 5a, the lobe ratios from constant frequency estimates exhibit a more pronounced

asymmetry. A fit is performed over constant wavenumber lobe ratios (full markers) for which $4 < k/k_p < 22$, providing the parametrization:

$$r_i = 0.34 k/k_p - 0.45. \tag{21}$$

Above $k/k_p \simeq 22$, the lobe ratios progressively decrease, except if the background term $C^{\text{st}} > 0$ is removed (transparent markers on Fig. 5). The lobe ratio decrease is natural since the lobe ratios without background are:

$$r_i' = \frac{E_i - C^{\text{st}}}{E_0 - C^{\text{st}}} > r_i \tag{22}$$

as long as $r_i > 1$ and the proportion of background noise increases towards shorter scales. We cannot however formally associate this noise with an actual surface waves signal.

The Stokes drift current for linear waves in deep water is (Kenyon, 1969)

$$u_s(z) = \int\limits_0^\infty dk\, V(k)\, m_1(k)\, e^{2kz}, \tag{23}$$

where

$$V(k) = 2\sigma(k)\, k E_{\text{free}}(k) \tag{24}$$

is plotted on Fig. 6a, and where the impact of the wave field directionality is included in the factor

$$m_1(k) = \sqrt{a_1^2 + b_1^2}, \tag{25}$$





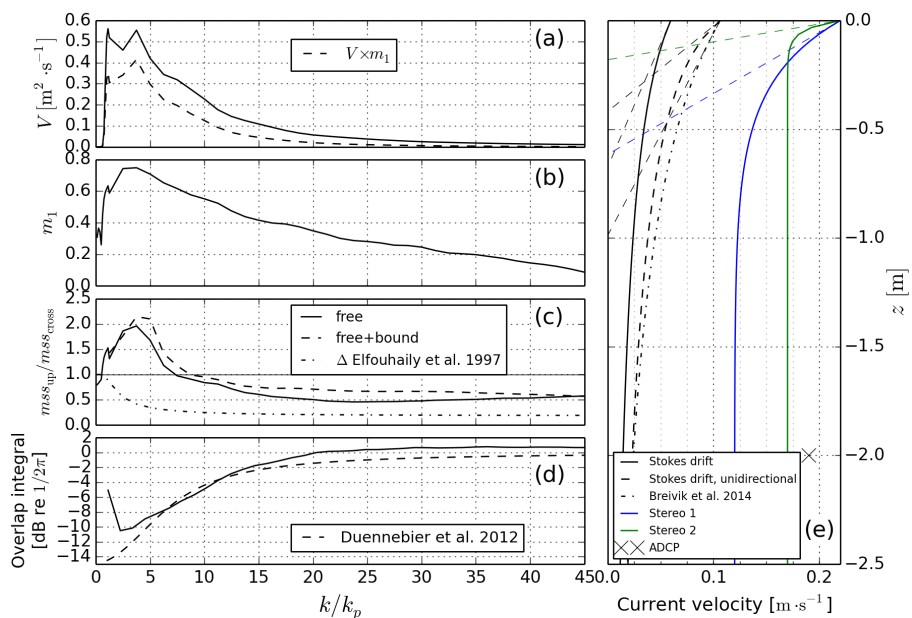

**Figure 6.** Short waves contribution to various sea state variables. (a): Spectrum of non-directional Stokes drift, Eq. (24). (b): Stokes drift directional correction, Eq. (23). (c): mean square slopes upwind over crosswind ratio, Eq. (30) and (31). (d): overlap integral. (e): near-surface current profiles (see Appendix), with the two parameters exponential integral profile of Breivik et al. (2014, their Eq. (16)).

plotted on Fig. 6b with

$$a_1(k) = \int_0^{2\pi} d\theta \, M_{\text{free}}(k,\theta) \cos\left(\theta - \bar{\theta}\right) \tag{26}$$

and

$$b_1(k) = \int_0^{2\pi} d\theta \, M_{\text{free}}(k,\theta) \sin\left(\theta - \bar{\theta}\right) \tag{27}$$

5  the Longuet-Higgins coefficients with respect to the mean wave propagation direction, where

$$E(k,\theta) = M(k,\theta) E(k) \tag{28}$$

and

$$\int_0^{2\pi} M(k,\theta) \, d\theta = 1. \tag{29}$$



The resulting Stokes drift vertical profile has been plotted on Fig. 6e, together with two profiles compatible with the effective current measured from stereo-video (see Appendix). Waves slightly shorter than peak waves are the main contributors to the Stokes drift (Fig. 6a). Half of the Stokes drift is carried by waves with frequencies greater than $0.4\,\mathrm{Hz}$ approximately (wavelength $10\,\mathrm{m}$). In order to correct for the stereo device field of view limitation (long waves are indeed not spatially resolved), the wavenumber spectrum for $k < \Delta k$ has been evaluated from their frequency spectrum using using the jacobian transform. In particular, the short waves bimodality substantially reduces their contribution to the Stokes drift. At a given wave scale, contributions symmetric with respect to the mean wave propagation direction cancel out laterally, resulting in a decrease of the Stokes drift at those scales (Fig. 6b). In particular, the Stokes drift at $z = 0$ is reduced by $44\%$ (from $0.11$ to $0.06\,\mathrm{m\cdot s^{-1}}$), which is greater than the approximately $20\%$ reduction reported in Ardhuin et al. (2009) and Breivik et al. (2014). Mean square slopes in the upwind and cross-wind direction are defined by

$$mss_{\mathrm{up}}(k) = \int_0^{2\pi} d\theta\, k^2 E(k,\theta) \cos^2\left(\theta - \bar{\theta}\right) \tag{30}$$

and

$$mss_{\mathrm{cross}}(k) = \int_0^{2\pi} d\theta\, k^2 E(k,\theta) \sin^2\left(\theta - \bar{\theta}\right), \tag{31}$$

and are of particular interest for ocean remote sensing (Munk, 2009). Due to the wave field bimodality, the mean square slope is rather carried by cross wind propagating waves than upwind (Fig. 6c) at short scales, as it was qualitatively described in Elfouhaily's delta ratio (Elfouhaily et al., 1997). Bound waves cause a slight increase of the mean square-slopes in the upwind direction. Finally, short waves directional distributions are critical in understanding the source of seismo-acoustic noise (Farrell and Munk, 2010), occasioned by quasi-stationnary pressure oscillations at the sea surface (Longuet-Higgins, 1950). The spectrum of stationary pressure waves can be written as

$$F_p = F_{p,\mathrm{free}} + F_{p,\mathrm{bound}}, \tag{32}$$

where the free waves contribution is proportional to the overlap integral $I$ (Wilson et al., 2003)

$$F_{p,\mathrm{free}} \propto E_{\mathrm{free}}^2(k) I(k), \tag{33}$$

given by

$$I(k) = \frac{2 \int_0^{\pi} d\theta\, E_{\mathrm{free}}(k,\theta) E_{\mathrm{free}}(k,\theta+\pi)}{E_{\mathrm{free}}^2(k)}. \tag{34}$$

The correction arising from bound harmonics $F_{p,\mathrm{bound}}$ has never been rigorously considered in past studies but should remain weak. The overlap integral (34) has been plotted on Fig. 6d. For the same energy level at given wave scale, the overlap integral is increased from a unimodal to a bimodal directional distribution. In particular, at short enough scales, more energy should be radiated by a bimodal surface wave field than by an equivalent isotropic wave field (for which the value $1/(2\pi)$ is reached). The parametrization of Duennebier et al. (2012) is also superimposed.





## 5 Discussion and summary

The characteristics of a bimodal short surface waves energy distribution are extracted from the spectrum of a single stereo-video reconstruction of the sea-surface at the Acqua Alta platform. Peak positions and lobe ratios are computed which can quantitatively summarize the observations, with associated parameterizations.

The domain of surface waves which can be measured with this system depends on the configuration of the device. Stereo video has a wide scale coverage and an upper-bound that is not limited by the Nyquist frequency and wavelength (here $f_s/2 = 7.5\,\mathrm{Hz}$ and $1/(2\Delta x) = 15.7\,\mathrm{rad}\cdot\mathrm{m}^{-1}$), but rather by the accuracy of reconstruction of short waves of small amplitudes. The effective directional resolution can be computed using

$$\Delta\theta \sim \arctan\left[\frac{\max(\Delta k_x, \Delta k_y)}{\kappa(f, U(f))}\right]. \tag{35}$$

In our case, for $1\,\mathrm{Hz}$ waves $\Delta\theta \sim 5°$, and for $0.5\,\mathrm{Hz}$ waves, it reaches $15°$.

    Bimodality has been characterized by extracting the positions of the three main points from directional distributions of the free waves, either at constant frequency or constant wavenumber (see Fig. 4). Free waves only are affected by bimodality both at given wavenumber and frequency. Moreover, bound waves distribution can be deduced from the one of free wavess (Leckler et al., 2015). The short waves field bimodality starts growing between $k/k_p = 3.6$ and $k/k_p = 4.3$ from constant wavenumber

snapshots, or between $k/k_p = 4.8$ ($f/f_p = 2.16$) and $k/k_p = 5.2$ ($f/f_p = 2.23$) from constant frequency snapshots. Bimodality may be initiated at even larger scales and not detected, due to a directional resolution at those scales which is smaller than the peak distance - Eq. (35). The two peaks then detach from the main direction $\theta_m$. Apart from the asymmetry introduced by the current, the three points characterizing bimodality sensibly fluctuate around their positions, reaching a distance of $\sim 160°$ towards $k/k_p = 45$, the latter being the accepted limit for stereo-video measurements validity. The real bimodal directional

distribution differs from its parameterizations mainly at wave scales smaller than $k/k_p = 22$. This is particularly the case for the peak furthest from the current direction at given frequency (see arrow on Fig. 5a) which gets away from the parametrization by slowly moving closer to the center of the directional distribution, while the constant wave number estimates remain close to the parametrization, with an almost perfectly symmetric distribution. This difference might come from the effect of the current. Indeed, the two peaks at given wavenumber do not appear at the same frequency because of the presence of the current. For

example, on Fig. 4d, the waves at $k = 4.0\,\mathrm{rad}\cdot\mathrm{m}^{-1}$ exhibit a bimodal behavior which is symmetric with respect to the main wave propagation direction $\theta_m$. In the absence of current, the two peaks would appear at the same frequency $f = 1.0\,\mathrm{Hz}$. On this snapshot, the peak furthest from the current, $i.e.$ $\theta_1$, is located at a frequency $f = 0.95\,\mathrm{Hz}$, while the other peak is located at a frequency $f = 1.022\,\mathrm{Hz}$. The shift is larger for the former, $\theta_1$, than for $\theta_2$, hence the current is a cause of asymmetry in bimodality characteristics. As a consequence, the wavenumber parametrization, is more robust against currents than is the

frequency one, as was already observed by Wyatt (2012). This is the one chosen throughout this paper. This asymmetry is also visible on Fig. 5b.

    We have reported on new stereo-video recordings of ocean waves, that offer a wider range of resolved scales than previous datasets, up to $k/k_p = 45$. Looking at free waves, the bimodal nature of their directional distribution is more pronounced at the shorter scales, with a separation of the two peaks that exceeds $160°$. This distribution was found to reduce the Stokes drift by





over $40\,\%$ compared to a unidirectional wave field, with a significant source of acoustic noise due to waves in opposing directions, typically larger than an isotropic spectrum for $k/k_p > 20$. These effects are partly compensated for by the importance of bound harmonics which have directions closer to the mean wave direction. The analysis of the contribution of these nonlinear components to the Stokes drift and acoustic noise is beyond the scope of the present paper.

## 5   Appendix A: Free waves extraction and current vector estimation

The extraction of free waves components from the surface elevation spectrum is here detailed. Looking at snapshots of Fig. 2 and 3, there is no ambiguity on the distinction between free (along the dispersion line, black) and bound waves (white line), except if the spatial resolution is limiting. From Eq. (4) to (6), their location in the $(\boldsymbol{k}, f)$ space is determined by the value of the effective current $\boldsymbol{U}$, Eq. (6), at each wave scale. It depends on the true near-surface current vertical profile $u(z)$.

### 10   A1   Effective current measurement

Starting from a snapshot of the surface elevation spectrum at given frequency or wavenumber (see Fig. 4 for example), the estimate of the effective current which minimizes the cost function

$$g\left(U_x, U_y\right) = \sum_{j=1}^{N} \frac{w_j}{\sigma^2} \left[2\pi f_j - \sqrt{gk_j} - k_j \left(U_x \cos\theta_j + U_y \sin\theta_j\right)\right]^2 \tag{A1}$$

is retained, where $(U_x, U_y)$ are the coordinates of the effective current vector in the local frame and $w_j$ are empirical weights, normalized so that $\sum_{j=1}^{N} w_j = N$. The flexibility of this method relies upon a careful choice of data points and weights.

   This choice is here exposed for the case of a constant frequency snapshot. First, a rough estimate of the current is required in order to approximately locate the dispersion relation. For this experiment, the current vector does not much vary with wave scales. This estimate is obtained by manually selecting data points on the dispersion relation of $\sim 1\,\mathrm{Hz}$ waves (ten are enough) and by minimizing the cost function with equal weights. The index 0 is put on the current value obtained ($U_0 = 0.22\,\mathrm{m\cdot s^{-1}}$ and $\alpha_0 = 102°$). A second function cost function is computed by keeping only data points for which

$$\kappa(\boldsymbol{f}_j, \boldsymbol{U}_0) - 0.1\kappa(\boldsymbol{f}_j) < k_j < \kappa(\boldsymbol{f}_j, \boldsymbol{U}_0) + 0.1\kappa(\boldsymbol{f}_j). \tag{A2}$$

Then, among the rest, the ones with the lowest signal to noise ratio are removed

$$\frac{E_j}{\max_j(E_j)} > 0.01 \tag{A3}$$

The weights are

$$w_j \propto \frac{E_j - \min_j(E_j)}{\max_j(E_j)} dS_j \tag{A4}$$

then normalized, where the index $j$ runs over remaining data points, and $dS_j$ stands for the elementary surface around data point $j$. The minimization algorithm is initiated with values $U_0$ and $\alpha_0$, and run until convergence at each frequency, providing





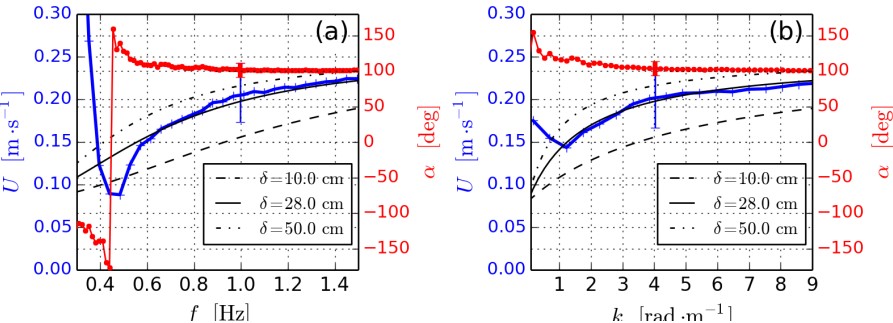

**Figure A1.** Effective current magnitude (blue) and direction (red) as a function of frequency and wavenumber, after smoothing over 5 adjacent points. Superimposed are the analytical profiles corresponding to a typical wind drift current when $u_b = 10\,\mathrm{cm \cdot s^{-1}}$ with various values of $\delta$ (see text for explanations).

a more accurate result than the rough estimate. Finally, free waves are isolated using this more accurate estimate, which are the points checking

$$\kappa(\boldsymbol{f}_j, \boldsymbol{U})/1.15 < k_j < 1.15\kappa(\boldsymbol{f}_j, \boldsymbol{U}) \tag{A5}$$

The same procedure can be applied to constant wavenumber snapshots. It is the same as the one of previous paragraph, after
having exchanged $k$ with $f$ and $\kappa$ with $\omega/(2\pi)$.

## A1 Current profile

The effective current values at all wave scales from the extraction of free waves are plotted on Fig. A1. Either as a function of frequency or wavenumber, both estimates show a gradual increase of the effective current magnitude towards $U_0 = 0.22\,\mathrm{m \cdot s^{-1}}$ and $\alpha_0 = 102°$. These values are in agreement with ADCP measurements indicating a current of $0.19\,\mathrm{m \cdot s^{-1}}$ flowing from the
direction $110°$ at $2\,\mathrm{m}$ below the surface, which is already too deep to significantly influence the effective current. Effective current for typical wind drift profiles $u(z) = u_a + u_b e^{z/\delta}$ are plotted on Fig. A1 for various values of $\delta$. We assume that $u_b = 0.1\,\mathrm{m \cdot s^{-1}}$, *i.e.* $1\,\%$ of the surface wind speed, and $u_a = U_0 - u_b = 0.12\,\mathrm{m \cdot s^{-1}}$, yielding a surface vertical shear $u_b/\delta = 0.36\,\mathrm{s^{-1}}$. Two plausible profiles have been plotted on Fig. 6e, for which

$$u(z)[\mathrm{m \cdot s^{-1}}] = 0.12 + 0.1e^{z/0.28[\mathrm{m}]} \tag{A1}$$

denoted Stereo 1, and

$$u(z)[\mathrm{m \cdot s^{-1}}] = 0.17 + 0.05e^{z/0.04[\mathrm{m}]} \tag{A2}$$

Stereo 2.

*Competing interests.* The authors declare that they have no conflict of interest.



*Acknowledgements.* This work is supported by LabexMer via grant ANR-10-LABX-19-01, and Copernicus Marine Environment Monitoring Service (CMEMS) as part of the Service Evolution program. Installation of the stereo system was supported by the funding from the Flagship Project RITMARE. The Italian Research for the Sea coordinated by the Italian National Research Council and funded by the Italian Ministry of Education, University and Research within the National Research Program 2011-2015.





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
