# Peer review of "Note on the directional properties of meter-scale gravity waves"

_Ocean Science, 2017_

## Referee Comment (RC1) · Anonymous Referee #1 · 22 Aug 2017

The manuscript describes an interesting analysis showing spectral bimodality at high wavenumbers (frequencies). This topic has been studies extensively, both numerically and with the aid of field observations, over the past years. An interesting and original conclusion of the present manuscript is the effect that current exerts on the asymmetry of the lobes. In my opinion, this manuscript is suitable for the Journal of Ocean Science. However, there are a number of comments and concerns that should be addressed by the Authors.

1) The manuscript seems to conclude that bound waves do not play a role on high wavenumber bimodality and this is consistent with a numerical investigation in Toffoli, A., M. Onorato, E. M. Bitner-Gregersen, and J. Monbaliu (2010), Development of a bimodal structure in ocean wave spectra, J. Geophys. Res., 115, C03006,

doi:10.1029/2009JC005495, where they showed that free wave nonlinearity is causing the bimodal lobes to form. However, the Authors mention in the abstract (line 2, page 1) that "distribution can be obscured by the presence of bound waves". Just looking at figure 4, this statement does not seem to be very relevant. What do the Authors actually mean with "obscured by the presence of bound waves"?

2) In general, the Introduction is a bit weak. There is an extensive literature on the high frequency bimodality of the wave spectrum, which could be discussed in mored details. In addition to field observations, there are a number of numerical investigations that shows and tries to explain the formation of this high frequency bimodality. Besides the cited Banner and Young (1994) and Gagnaire-Renou et al (2010), the Authors should refer to Alves, J. H. G. M., and M. L. Banner (2003), Performance of a saturation- based dissipation-rate source term in modeling the fetch-limited evolution of wind waves, J. Phys. Oceanogr., 33, 1274–1298; Dysthe, K. B., K. Trulsen, H. Krogstad, and H. Socquet-Juglard (2003), Evolution of a narrow-band spectrum of random surface gravity waves, J. Fluid Mech., 478, 1–10; and Toffoli, A., M. Onorato, E. M. Bitner-Gregersen, and J. Monbaliu (2010), Development of a bimodal structure in ocean wave spectra, J. Geophys. Res., 115, C03006, doi:10.1029/2009JC005495, among others.

3) Reference to Munk (2009) is not well discussed, I think, I do not quite understand what the "challenge" mentioned at line 17 of page 1 is.

4) At line 4, page 2, there is reference to the "directional distribution of backscatter". What does the backscatter refer to?

5) It seems that the main contribution of the manuscript is an extension of the work in Lecker et al (2005), but no specific details about this referred study are provided. Lecker et al (2005) should be discussed in more details to better highlight the novelty of the present manuscript.

6) It is mentioned that a Fourier analysis is conducted over a physical domain of dimension 25.6 X 25.6 m^2. This seems quite small to me, considering the the dominant wavelength is of about 45m. It means that physical domain does not contain one full wave form, creating uncertainties in the Fourier analysis. It is indeed a well known problem that the frequency domain is not well resolved if a whole number of periods is not present in the physical space. The resulting wave spectrum is therefore questionable. The Authors should make sure that their domain contains at least one full dominant wave period for the Fourier analysis.

7) At line 5, page 6, the difference interaction should be k = k1 - k2, right?

8) Not sure I understand the meaning of "background spectrum" at line 10 of page 10.

9) I don't think I understand the reasoning at the beginning of page 12 (around line 5). Also, what "their" refer to in "...the short wave bimodality substantially reduces their contribution to Stokes drift"?

---

## Referee Comment (RC2) · F. Dias (Referee) · 23 Aug 2017

The abstract is a bit technical and should be rewritten to emphasize the key finding of the note. The first sentence of the abstract is not very clear, in particular the expression "is bimodal for frequencies above twice the peak frequency". Maybe the definition of bimodality should be given, since it seems that not everybody uses the same definition in the literature.

The main question of the reviewer is: what is the point of removing bound harmonics? For example, Romero & Melville studied bimodality without removing any bound harmonics. Consequently, the second part of the first sentence of the abstract is a bit misleading.

[Figure]

Overall, I know that it is obvious for the authors but I am not sure that I always see exactly where the bimodality is present in the figures. For example, could the authors add some arrows in Figures 2 and 3, that match the text on Page 6, line 2 ("... detach from a main direction ...")

Page 1, last line: the bimodality is caused by the nonlinear cascade of wave energy from dominant to high frequencies. So not by free waves?

Page 2, sentence lines 1,2,3: I do not understand the sentence.

Page 2, last line: "increasing away from the cameras ..." – to the left or to the right?

Page 4, line 11: there is a mixture of vectors and scalars (at least in the notation). Same in equation (6).

Page 7, line 23 and Page 8, second line of Caption of Figure 4: \alpha seems to have two different meanings. Please change the notation.

Page 8, Figure 4(a): Should there be a subscript "free" instead of "bound"?

Page 8, lines 6 & 7: circles and disks should be reversed.

Page 9, line 1: I do not understand where the $k/k_p = 4$ comes from.

Page 10, equation (23): Should the integral be from $-\infty$ to 0 ?

Page 11, Figure 6: the caption refers to equations (30) and (31), but these equations come after the reference to figure 6 in the text.

Page 12, line 5: using repeated twice

Page 12, line 18: I could be wrong but I am not sure that "occasion" is a verb in English

Page 13, line 11: What are these three main points? (I am lost)

Page 13, line 13: waves

Appendix page 14, equation (A1): why is the dispersion relation that of deep water? Is

a factor 1/N missing?

Figure A1 page 15: clearly say that the difference between the two figures is f (left) vs k (right). (a) and (b) do not even appear in the caption.

Page 15, line 2: "points checking" ???

Page 15, line 12 and caption of figure A1: use the same units for the velocity u_b

---

## Author Comment (AC1) · 27 Sep 2017

The authors would like to thank the reviewers for their comments, which helped improving this manuscript. The questions asked by the reviewers are rewritten in bold and answers follow.

**1) The manuscript seems to conclude that bound waves do not play a role on high wavenumber bimodality and this is consistent with a numerical investigation in Toffoli, A., M. Onorato, E. M. Bitner-Gregersen, and J. Monbaliu (2010), Development of a bimodal structure in ocean wave spectra, J. Geophys. Res., 115, C03006, doi:10.1029/2009JC005495, where they showed that free wave nonlinearity is causing the bimodal lobes to form. However, the Authors mention in**

the abstract (line 2,page 1) that "distribution can be obscured by the presence of bound waves". Just looking at figure 4, this statement does not seem to be very relevant. What do the Authors actually mean with "obscured by the presence of bound waves"?

The role attributed to bound waves in the manuscript has been highlighted by both reviewers. This paragraph takes both remarks into account. The extraction of bound waves is one of the interesting features allowed by the stereo-video technique. Without anticipating any role played by bound waves in the origination of bimodality, the sentence "distribution can be obscured by the presence of bound waves", is probably inaccurate. It is indeed true from figure 4 that bound waves do not "obscur" bimodal directional distributions, i.e. make bimodal directional distributions look unimodal. However, looking at the same figure, the parameters of bimodality presented on figure 5 are strongly influenced by the presence of bound waves, particularly the lobe ratios, equation (20). In addition, the bound waves depend on the full spectrum of free waves (Hasselmann, 1962; Janssen, 2009). Removing these bound waves allows to get rid of the potential variability of the spectrum of free waves, especially the long waves part, which can be quite different from the short waves part. For these reasons, the extraction of bound waves is important for quantifying the lobe ratio and other parameters that define the spectral shape. We have thus changed the abstract, with the new sentence

*"The later indeed tend to reduce the contrast between the two peaks and the background".*

2) In general, the Introduction is a bit weak. There is an extensive literature on the high frequency bimodality of the wave spectrum, which could be discussed in mored details. In addition to field observations, there are a number of numerical investigations that shows and tries to explain the formation of this high frequency bimodality. Besides the cited Banner and Young (1994) and Gagnaire-Renou et al (2010), the Authors should refer to Alves, J. H. G. M., and M. L. Banner (2003), Performance of a saturation-based dissipation-rate source term in modeling the fetch-limited evolution of wind waves, J. Phys. Oceanogr., 33, 1274–1298; Dysthe, K. B., K. Trulsen, H. Krogstad, and H. Socquet-Juglard (2003), Evolution of a narrow-band spectrum of random surface gravity waves, J. Fluid Mech., 478, 1–10; and Toffoli, A., M. Onorato, E. M. Bitner-Gregersen, and J. Monbaliu (2010), Development of a bimodal structure in ocean wave spectra, J. Geophys. Res., 115, C03006, doi:10.1029/2009JC005495, among others.

References to the aforementioned articles have been added to the introduction :

*'"Bimodality is also found after having solved for the nonlinear evolution equation of the surface elevation field, whether computing it for gaussian wave packets according to a Nonlinear Schrödinger equation (Dysthe et al., 2013) or for unimodal wave spectra from the Euler equations (Toffoli et al., 2010)"*

**3) Reference to Munk (2009) is not well discussed, I think, I do not quite understand what the "challenge" mentioned at line 17 of page 1 is.**

The challenge here presented has to do with the reflectance measurements presented in Bréon and Henriot (2006). These measurements are integrated over wave scales and present puzzling simple relationships between wind speed and cross-wind or down-wind slope variance. However, this data gives no information on the underlying distribution among frequencies and wave numbers. The challenge here was to obtain frequency/wave number resolved data, to help understand the integrated relationship.

**4) At line 4, page 2, there is reference to the "directional distribution of backscatter". What does the backscatter refer to?**

The backscatter refers to the radar backscatter as a function of azimuth. This has been corrected in the introduction :

*"The distribution of radar backscatter as a function of azimuth clearly shows that the directional wave spectrum is unimodal above $6$ cm wavelength in the gravity-capillary range (see the review in Elfouhaily et al., 1997)"*

**5) It seems that the main contribution of the manuscript is an extension of the work in Lecker et al (2005), but no specific details about this referred study are provided. Lecker et al (2005) should be discussed in more details to better highlight the novelty of the present manuscript.**

The discussion on Leckler et al. (2015) has been expanded in the introduction :

*"As shown by Leckler et al. (2015), stereo-video imagery is capable of resolving these waves and provide information on the time and space scales needed to interpret integrated wave parameters such as surface slope. In this paper, a record from a young wind waves field taken from a platform in Crimea have been analyzed. In particular, the presence of harmonics, the shift induced by the current on the short surface waves dispersion relation and the wave field bimodality were part of the conclusions. Here we focus on the short waves field bimodality and extend their analysis to the whole range of frequencies. The characteristics of bimodality are here quantified and the consequences on physical variables detailed."*

**6) It is mentioned that a Fourier analysis is conducted over a physical domain of dimension 25.6 X 25.6 mËĘ2. This seems quite small to me, considering the the dominant wavelength is of about 45m. It means that physical domain does not contain one full wave form, creating uncertainties in the Fourier analysis. It is indeed a well known problem that the frequency domain is not well resolved if a whole number of periods is not present in the physical space. The resulting wave spectrum is therefore questionable. The Authors should make sure that their domain contains at least one full dominant wave period for the Fourier analysis.**

The limitation of having a small analysis window compared to the dominant wave spatial scales reflects on our inability to resolve properly wave numbers around the spectral peak (this fact is quite clear in Figure 3, with slices taken at constant frequencies). This problem does not present for wave periods, since the record duration is longer enough to contain hundreds wave periods. However, smaller scales are well resolved. More-

over, we have reduced the influence of aliasing errors accounting for a limited portion of the 3-D spectrum (Eqs. 11 trough 15).

This is a well-know issue in array processing for ocean waves (e.g. Kinsman, 1965; Donelan et al., 1985) or seismic waves. For wave components with wavelengths larger than the array size, the usual technique is that of a "slope array" (e.g. Graber et al., 2000) that gives a robust estimate of mean direction and spread (and at least first 5 moments of the directional distribution as given by a buoy). In Leckler et al. (2015) this array processing was combined with the direct FFT to give a full spectrum from the peak to the short waves because they computed the second order spectrum from the dominant waves. Here we focus on the shorter components.

We have thus added : *"Wavelengths longer than 25 m can be resolved using standard slope array techniques (e.g. Graber et al., 2000) as done by Leckler et al. (2015) and Benetazzo (2006). These longer components are not the focus of the present paper. "*

**7) At line 5, page 6, the difference interaction should be k = k1 - k2, right?**

The reviewer is correct, but (k1+k2,omega1-omega2) is the same as (k1-k2, omega1+omega2). We have modified the text to make it more intuitive:

*"The difference interaction gives $\vec{k} = \vec{k}_1 - \vec{k}_2$ and $\omega = \left| \omega(\vec{k}_1) + \omega(\vec{k}_2) \right|$, $E_{\mathrm{diff}}$."*

Moreover, the interaction kernel from Sharma and Dean (1979) uses the minus sign between the phases.

**8) Not sure I understand the meaning of "background spectrum" at line 10 of page 10.**

This sentence is probably misleading. The background spectrum is not another kind of spectrum. This name is probably inappropriate. The directional distribution at a given wave scale does not always fall down to zero. In fact, there is always a bit of energy remaining, especially towards smaller scales. The origin of this energy may be either

an actual surface waves signal or noise. This would require further investigations. This background is similar to the term $\alpha$ introduced by Tyler et al. (1974) in their fitting function to account for a non-zero energy level for waves propagating in opposite directions. This has been corrected in the manuscript :

*"We can note that they are particularly sensitive to the background energy level. This level is given by the constant term $C^{\mathrm{st}}$ of the fitting function (17), without knowning whether this level is actual surface waves signal or noise."*

**9) I don't think I understand the reasoning at the beginning of page 12 (around line 5). Also, what "their" refer to in " ... the short wave bimodality substantially reduces their contribution to Stokes drift"?**

Indeed "their" refers to the contribution of short waves. This has been corrected in the manuscript :

*" In particular, the short waves bimodality substantially reduces the contribution of those waves to the Stokes drift."*

**References**

Alves, J. H. G. and Banner, M. L.: Performance of a saturation-based dissipation rate source term in modeling the fetch-limited evolution of wind waves, J. Phys. Oceanogr., 33, 1274–1298, 10.1175/1520-0485(2003)033<1274:poasds>2.0.co;2, 2003.

Benetazzo, A.: Measurements of short water waves using stereo matched image sequences, Coastal. Eng., 53, 1013–1032, 2006.

Bréon, F. M. and Henriot, N.: Spaceborne observations of ocean glint reflectance and modeling of wave slope distributions, J. Geophys. Res., 111, 10.1029/2005jc003343, 2006.

Donelan, M. A., Hamilton, J. and Hui, W. H.: Directional Spectra of Wind-Generated Waves, Philos. Trans. Royal Soc. A, 315, 509–562, 10.1098/rsta.1985.0054, 1985.

Duennebier, F. K., Lukas, R., Nosal, E.-M., Aucan, J., and Weller, R. A.: Wind, Waves,

and Acoustic Background Levels at Station ALOHA, J. Geophys. Res., 117, C03:017, 10.1029/2011JC007267, 2012.

Dysthe, K. B., Trulsen, K., Krogstad, H. E. and Socquet-Juglard, H.: Evolution of a narrow-band spectrum of random surface gravity waves, J. Fluid Mech., 478, 10.1017/s0022112002002616, 2003.

Elfouhaily, T., Chapron, B., Katsaros, K., and Vandemark, D.: A unified directional spectrum for long and short wind-driven waves, J. of Geophys. Res., 102, 15:781–15:796, 1997.

Graber, H. C., Terray, E. A., Donelan, M. A., Drennan, W. M., Leer, J. C. V., and Peters, D. B.: ASIS—A New Air–Sea Interaction Spar Buoy: Design and Performance at Sea, J. Atmos. Oceanic Technol., 17, 708–720, 2000.

Hasselmann, K.: On the non-linear energy transfer in a gravity wave spectrum, part 1: general theory, J. Fluid Mech., 12, 481–501, 10.1017/s0022112062000373, 1962.

Janssen, P. A. E. M.: On some consequences of the canonical transformation in the Hamiltonian theory of water waves, J. Fluid Mech., 637, 1–44, 10.1017/s0022112009008131, 2009.

Kinsman, B: Wind Waves, Prentice-Hall, 1965.

Leckler, F., Ardhuin, F., Peureux, C., Benetazzo, A., Bergamasco, F., and Dulov, V.: Analysis and interpretation of frequency-wavenumber spectra of young wind waves, J. Phys. Oceanogr., 45, 2484–2496, 10.1175/JPO-D-14-0237.1, 2015.

Munk, W.: An Inconvenient Sea Truth: Spread, Steepness, and Skewness of Surface Slopes, Annu. Rev. Mar. Sci., 1, 377–415, n10.1146/annurev.marine.010908.163940, 2009.

Sharma, J. N. and Dean, R. G.:Development and evaluation of a procedure for simulating a random directional second order sea surface and associated wave forces, University of Delaware, 1979.

Toffoli, A., Onorato, M., Bitner-Gregersen, E. M. and Monbaliu, J.: Development of a bimodal structure in ocean wave spectra, J. Geophys. Res., 115, C03:006, doi:10.1029/2009JC005495, 2010.

Tyler, G., Teague, C., Stewart, R., Peterson, A., Munk, W. and Joy, J.: Wave directional spectra from synthetic aperture observations of radio scatter, Deep Sea Research and Oceanographic Abstracts,21, 989–1016, 10.1016/0011-7471(74)90063-1, 1974.
* * *

---

## Author Comment (AC2) · 27 Sep 2017

The authors would like to thank the reviewers for their comments, which helped improving this manuscript. The questions asked by the reviewers are rewritten in bold and answers follow.

**The abstract is a bit technical and should be rewritten to emphasize the key finding of the note. The first sentence of the abstract is not very clear, in particular the expression "is bimodal for frequencies above twice the peak frequency". Maybe the definition of bimodality should be given, since it seems that not everybody uses the same definition in the literature.**

We have now included a definition of bimodality in the abstract :

[Figure]

*"their directional distribution exhibits two peaks in different directions and a minimum between."*

The key finding of the note is summarized by the last sentence of the abstract:

*"These observations extend to shorter components previous measurements, and have important consequences for wave properties sensitive to the directional distribution, such as surface slopes, Stokes drift or microseism sources."*

**The main question of the reviewer is: what is the point of removing bound harmonics? For example, Romero Melville studied bimodality without removing any bound harmonics. Consequently, the second part of the first sentence of the abstract is a bit misleading.**

Please refer to the reply to reviewer 1, question 1.

**Overall, I know that it is obvious for the authors but I am not sure that I always see exactly where the bimodality is present in the figures. For example, could the authors add some arrows in Figures 2 and 3, that match the text on Page 6, line 2 (". . . detach from a main direction . . .")**

Arrows have been added on figure 3 (page 5) to locate the directions of the two lobes and of the "main direction".

**Page 1, last line: the bimodality is caused by the nonlinear cascade of wave energy from dominant to high frequencies. So not by free waves?**

In a weakly turbulent framework, the nonlinear energy cascade involves waves from the first and the third (and higher) order of nonlinearity. Bound waves also result from nonlinearities, but it is known from Hasselmann (1962) that non stationary energy transfer occurs among free waves (Snl source term). This same term has been confirmed to be a source of bimodality by the references cited in the introduction.

**Page 2, sentence lines 1,2,3: I do not understand the sentence.**

[Figure]

This has been corrected in the manuscript :

*"The model results of Gagnaire-Renou et al. (2010, their figure 18) show that bimodality is followed at smaller scales by a return to a unimodal directional distribution, somewhere below $f/f_p = 10$, depending on the parameterizations of wave generation and dissipation."*

**Page 2, last line: "increasing away from the cameras . . ." – to the left or to the right?**

The sentence is probably misleading. For clarity, the camera look direction has been added to figure 1 (page 4), so that the readers can easily figure the way cameras are looking.

**Page 4, line 11: there is a mixture of vectors and scalars (at least in the notation). Same in equation (6).**

Notations have been changed in equation (4).

**Page 7, line 23 and Page 8, second line of Caption of Figure 4: $\alpha$ seems to have two different meanings. Please change the notation.**

This has been corrected in the definition of the fitting function, equation (17) and (18).

**Page 8, Figure 4(a): Should there be a subscript "free" instead of "bound"?**

This has been corrected in the manuscript. The caption of this figure has been modified as well in order to make this thing clear : the spectrum of free waves is located between the 2 black solid lines.

**Page 8, lines 6  7: circles and disks should be reversed.**

This has been corrected in the manuscript :

*"Full markers (triangles, disks and stars) correspond to estimates from constant wavenumber snapshots while empty markers (circles, diamonds and upside down tri-*

*angles) correspond to estimates from constant frequency snapshots."*

**Page 9, line 1: I do not understand where the $k/k_p = 4$ comes from.**

The authors wanted to provide a lower bound for the appearance of bimodality. This value actually results from a mistake in the computations. Looking at the directional distributions as a function of frequency, bimodality appears between f=0.410 and f=0.425 Hz. Concerning the directional distributions as a function of wave number, they appear between k=0.52 and 0.70 rad/m. The peak frequency being fp = 0.189 Hz, the corresponding wave number is kp = 0.146 rad/m (with a water depth of 17 m). In other words, the bounds for bimodality correspond to : 2.2<f/fp<2.3 or in terms of wave number 4.7<k/kp<5 3.6<k/kp<4.8 As the accuracy of these results is quite unknown, the value k/kp=5 could be retained as representative, rather than 4. In the previous estimate, the finite water depth had not been taken into account. This has been corrected on the parametrization of figure 5 and in the manuscript :

*"Bimodal profiles are first detected at $f = 0.43$ Hz and $k = 0.7$ rad $\cdot$ m$^{-1}$, corresponding approximately to $k/k_p = 5$."*

The parameter a=0.039 has not been affected by this change.

**Page 10, equation (23): Should the integral be from $-\infty$ to 0 ?**

The integral in equation (23) is performed over the wave number domain in order to account for the effect of waves of all scales. As a consequence, the Stokes drift has to be integrated across all wave scales, from 0 to infinity.

**Page 11, Figure 6: the caption refers to equations (30) and (31), but these equations come after the reference to figure 6 in the text.**

Figure 6 shows variables referenced in the text at successive locations. For convenience, they have been plotted on the same figure. Figure 6 has been shifted to the end of section 4.

**Page 12, line 5: using repeated twice**

This has been corrected in the manuscript accordingly.

**Page 12, line 18: I could be wrong but I am not sure that "occasion" is a verb in English**

This verb has been replaced by "caused" in the text,.

**Page 13, line 11: What are these three main points? (I am lost)**

A description has been added in the text :

*"Bimodality has been characterized by extracting the positions of the two bimodal peaks and the central minimum from directional distributions of the free waves, either at constant frequency or constant wavenumber (see Fig. 4)."*

**Page 13, line 13: waves**

This has been corrected in the text accordingly.

**Appendix page 14, equation (A1): why is the dispersion relation that of deep water? Is factor 1/N missing?**

The more general dispersion relation could be used as well, but its use only introduces insignificant differences for the domain of waves (typically above 1-2 times the peak frequency) which are analyzed by this algorithm. For example, at 0.189 Hz, the inversion of the dispersion relation in deep water only introduces errors of about 1The cost function has been adapted from Senet et al. (2001), equation (7), with the introduction of weights. The factor 1/N could be added but does not change the set of (Ux,Uy) which minimizes the cost function. However, the authors realized that the definition of factor $\sigma$ (renamed $\chi$) was missing. This has been added in the text, equation (A2).

**Figure A1 page 15: clearly say that the difference between the two figures is f (left) vs k (right). (a) and (b) do not even appear in the caption.**

This has been corrected in the caption of figure A1.

**Page 15, line 2: "points checking" ???**

This has been corrected in the manuscript :

*"Only the points with coordinates $(k_j, \theta_j)$ are kept if they fall in the interval [...] "*

**Page 15, line 12 and caption of figure A1: use the same units for the velocity $u_b$**

This has been corrected in the manuscript, caption of figure A1.

**References**

Gagnaire-Renou, E., Benoit, M., and Forget, P.: Ocean wave spectrum properties as derived from quasi-exact computations of nonlinear wave-wave interactions, J. Geophys. Res., 115, C12:058, 10.1029/2009JC005665, 2010.

Hasselmann, K.: On the non-linear energy transfer in a gravity wave spectrum, part 1: general theory, J. Fluid Mech., 12, 481–501, 10.1017/s0022112062000373, 1962.

Senet, C. M., Seemann, J., and Zeimer, F.: The near-surface current velocity determined from image sequences of the sea surface, IEEE Trans. Geosci. Remote Sensing, 39, 492–505, 10.1109/36.911108, 2001.

---

## Author Response (AR2)

**Answer to Referees (second review)**

*C. Peureux et al. 2017*

**Answer to Referee #1**

**1) The authors tried to explain the challenge mentioned in Munk (2009). However, this is not yet very clear to me. The authors mention some reflectance measurements presented in Breon and Henriot. What are these measurements about? I also noted that the author made no attempt to update the manuscript.**

Those reflectance measurements are related to the ocean waves mean square slopes. The manuscript has been corrected accordingly :

"In a landmark paper,  Munk (2009) analyzed the linear trends of down-wind and cross-wind mean square slopes of the sea surface, as measured by satellites (Breon and Henriot 2006). These trends cannot be explained by today's understanding of ocean wave spectra, and he proposed that there may be localized sources that could generate oblique propagating waves looking like with ship wakes . However, as he put it, the dataset says nothing about ..."

**2) As mentioned earlier, it seems that the manuscript leverages on Leckert et al (2005), but their results are not explained properly. The authors cannot expect the reader to study Leckler et al. (2005) before reading their manuscript. From the marked version is also quite clear that there is not additional explanation to explain Leckler et al. (2005), despite additional discussion being promised in the rebuttal. The only additional text refer to the contents of the manuscript.**

We have added some description of that earlier work:

"As shown by Leckler et al. (2015), stereo-video imagery is capable of resolving these waves and provide information on the time and space scales needed to interpret integrated wave parameters such as downwind and cross-wind mean square slopes. In that earlier paper, a record with young wind waves was analyzed ($U23=13.2m/s$, $f\_p = 0.33$). That record revealed the presence of second order harmonics, and a strong bimodality of the directional distribution. Here we use the same measurement method and analyse the directional properties of the free waves in more detail. In particular we analyze new data that provide a wider range of frequencies and quantitatively characterize the bimodality characteristics together its impacts on several physical variables."

**3) the difference contribution generates subharmonics, the frequency of which is $w_{diff} = \Delta(w)$ and $k_{diff} = \Delta k$. I do not understand why the authors says that the difference wave number is indeed a difference while the difference frequency is a sum. Can the authors be more specific? (note that psi_i - psi_j in second order, means $k_i X - w_i t - k_j + w_j t$; this means $k_i - k_j$ and $w_j - w_i$)**

This is correct. This has been corrected in the manuscript :
"$|w_i + w_j|$" → "$|w_i - w_j|$"

**4) page 4, line 18, what do the question marks refer to?**

It was a latex compilation error (missing reference) which has been corrected in the manuscript.

**List of changes**

Most changes have already been listed in the answer to the reviewers. Additional corrections are listed below, indexed by the line number of the revised manuscript.

> 1. The delta ratio of Elfouhaily et al. 1997 has been replaced by the estimate of mss_up/mss_cross computed from that delta ratio and the shape of the directional distribution given by Elfouhaily et al., namely $(2 + \text{Delta})/(2 - \text{Delta})$.

[revised manuscript text omitted]